# The Willingness to Pay for a Hypothetical Vaccine for the Coronavirus Disease 2019 (COVID-19)

**DOI:** 10.3390/ijerph182312450

**Published:** 2021-11-26

**Authors:** Yoshiro Tsutsui, Shosh Shahrabani, Eiji Yamamura, Ryohei Hayashi, Youki Kohsaka, Fumio Ohtake

**Affiliations:** 1Faculty of Social Relations, Kyoto Bunkyo University, Uji 611-0041, Japan; 2Head of Research Authority, The Max Stern Yezreel Valley College, Emek Yezreel P.O. 1930600, Israel; shoshs@gmail.com; 3Department of Economics, Seinan Gakuin University, Fukuoka 814-8511, Japan; yamaei@seinan-gu.ac.jp; 4School of Economics & Management, Kochi University of Technology, Kochi 780-8515, Japan; pey@ryohei.info; 5Department of Management Information, Kyoto College of Economics, Kyoto 610-1195, Japan; kohsaka@kyoto-econ.ac.jp; 6Center for Infectious Disease Education and Research, Osaka University, Osaka 560-0043, Japan; ohtake@econ.osaka-u.ac.jp

**Keywords:** COVID-19, vaccine, health belief model, risk attitude

## Abstract

This study investigates how people in Japan perceived the severity of and probability of infection from coronavirus disease 2019 (COVID-19), and how their willingness to purchase a hypothetical vaccine depends on these perceptions and their risk attitudes. We conducted a large-scale panel survey three times between 13 March to 13 April 2020 in Japan. By analyzing the data, we found that the perception of COVID-19 became more serious. The estimation of the fixed effect model reveals that a person becomes more willing to pay for a vaccine as the person evaluates COVID-19 as a more severe disease, considers a higher probability of infection, and becomes more risk averse. Since the sensitivity of willingness to pay for the vaccine on risk aversion increased during the period, the change in risk attitude contributed to an increase in willingness through the sensitivity channel, while it decreased through the magnitude channel.

## 1. Introduction

How do people in Japan practice prevention measures to protect themselves from the risks of coronavirus disease 2019 (COVID-19)? Do they rationally make decisions about prevention measures? To combat COVID-19, whether people practice appropriate preventive behavior is the key to not only protecting themselves, but also to preventing the spread of the disease. Since the externality of preventive behavior is large, governments in many countries locked down cities and shut down stores and offices. Many studies have been conducted to seek effective prevention measures against COVID-19 [1,2,3,4,5]. In the field of economics, research on quarantine [6,7,8] and prevention measures [9,10] has been carried out.

In Japan, there is no legislation to force people to self-isolate in their homes or offices. Therefore, the Japanese government requests the cooperation of citizens and applies moral pressure on them, relying on altruism and conformity to the social norm. To fulfill the request of voluntary self-control for preventive purposes, individuals are required to make a rational plan. This study investigates whether residents in Japan make rational preventive behavioral decisions.

Rational people will consider the benefits and costs associated with the COVID-19 in their decisions about adopting prevention behaviors. Benefits depend on how severe the disease is when one is infected, the probability of an infection, and the effectiveness of prevention measures, while costs refer to monetary and opportunity costs as well as the discomfort caused by the measures. These considerations were formalized as the health belief model (HBM), which has been one of the most extensively used theories to explain preventive health behavior in terms of certain belief patterns [11,12,13]. This model has been adapted to examine various health behaviors, including vaccination [14,15,16,17,18,19]. The HBM comprises several main categories that were identified as predictive factors of decisions regarding the influenza vaccine: perceived susceptibility, perceived severity, perceived benefits, perceived barriers, self-efficacy, and cues to action [12]. For example, several studies, such as [14,15], found the HBM categories among the predictive factors of the decision to receive a flu shot, such as higher levels of perceived seriousness of the illness and higher levels of perceived susceptibility (higher probability of contracting the disease). In addition, a large meta-analysis study indicates that risk likelihood, susceptibility, and severity significantly predicted vaccination behavior [20] (18 Brewer et al., 2007).

Numerous studies have been undertaken about vaccination against COVID-19 and the health belief model in the COVID-19 pandemic. For example, many papers reported the willingness to pay (WTP) for a hypothetical vaccine against COVID-19. Using an online survey answered by 566 individuals conducted between 18 April and 5 May 2020, in Chile, ref. [21] found that the individual’s willingness to pay (WTP) for a hypothetical COVID-19 vaccine was US $184. Similarly, using the data of 531 individuals conducted in July–August 2020, ref. [22] found that WTP of a vaccine was US $232 per vaccine, which is quite high in Chile. Using an online survey conducted from 2 April to 7 April 2020, in Ecuador, ref. [23] found that the average WTP values ranged from US $148 to 197. Ref. [24] conducted an online discrete choice experiment survey between June and July 2020, in China. They asked about a hypothetical vaccine and found that there was strong public preference for high effectiveness of the vaccine, followed by long protective duration, very few adverse events and being manufactured overseas. Price was the least important attribute. Using a cross-sectional survey conducted from 3 to 12 April 2020, in Malaysia, ref. [25] found that WTP was US $30.66 ± 18.12. In addition, they found respondents believe that the vaccination decreases the chance of infection. These studies asked about a hypothetical vaccine like the current study, and they found WTP was quite high. In addition, the conclusion of ref. [25] that HBM was supported is also consistent with the current study.

In contrast, those which studied these topics in Japan are relatively few. Some papers simply reported how many people showed the willingness to take a hypothetical vaccine using a cross-sectional survey [26,27,28] assessed the intention to be vaccinated for influenza and rubella, providing information about severe risks and susceptibility. However, the effect of the information was not necessarily clear-cut. In addition, they concluded that older individuals demonstrated vaccine hesitancy for both vaccinations, which seems opposite to the fact of the COVID-19 cases. Ref. [29], using a cross sectional survey, reported that perceived risks of COVID-19 and those of vaccine are associated with the willingness to take vaccine, being consistent with HBM. Using a cross sectional survey results, ref. [30] examined the psychological factors that predict staying at home during the COVID-19 pandemic. They found that perceived severity and self-efficacy significantly predicted greater levels of staying at home.

The following is not related to HBM nor vaccination, including two papers by economists. Using official labor data, ref. [31] documented heterogeneous changes in employment and earnings in response to the COVID-19 shocks. They reported that contingent workers are hit harder than regular workers, younger workers than older workers, females than males, and workers engaged in social and non-flexible jobs than those in ordinary and flexible jobs. Using prefecture-level daily data that contain the numbers of infectious and recovered people, ref. [32] calibrated the SIR-Macro model to find that a voluntary lockdown and a request-based lockdown play an important role in the low proportion of infectious individuals and the large decrease in consumption in Japan. Ref. [33] reported suicide cases in 2020 in Japan have increased from late July to November especially for women. Investigating the effects of weather, population and host factors on the outcome of COVID-19 in Japan, ref. [34] found that the strongest correlation was detected between fatalities and population density followed by total population and by humidity. Using a chatbot-based healthcare system named COOPERA (COVID-19: Operation for Personalized Empowerment to Render Smart Prevention and AN Care Seeking), ref. [35] analyzed 353,010 participants from Tokyo recruited from 27 March to 6 April 2020. They found that 95.6% of participants had no subjective symptoms. Ref. [36] found that a substantial proportion of older family carers had relatively low vaccine literacy.

Given the scarcity of vaccination and HBM study in Japan, in this study we comply with the HBM and formalize the model as follows: first, we introduce a hypothetical vaccine, which is expensive but effective, and investigate respondents’ willingness to purchase the vaccine. Since the effectiveness and the cost are provided here, rational models including the HBM, predict that the severity of the disease and the probability of infection are the main predicting factors. Even though in reality various preferences and attributes could play a role in addition to these basic factors, we focus on these two factors in this study by estimating the fixed effect model (FE). In addition, we estimate an FE model by incorporating several time-variant variables, which we conducted in three waves, to check the robustness of the results.

We conducted a panel survey from 13 March to 13 April 2020, which determined the willingness to pay for a hypothetical vaccine, how individuals evaluate the severity of COVID-19, how they predict the probability of infection, and their risk attitudes. The survey collected panel data during three waves, and enabled us to analyze how Japanese prevention behavior changed with the rapid spread of COVID-19.

Whereas most previous studies are based on cross-sectional survey, a merit of the current study is that it is based on a panel survey of larger number of respondents (around 4000), which enabled us to estimate a FE model to investigate the effect of the within-individual changes in the subjective probability of infection and severity of symptom when infected on the willingness to take a vaccine.

The remainder of this paper is organized as follows. In Section 2, we explain the survey, the COVID-19 situation during the observed period, our hypotheses, and the model tested. In addition to the baseline model, which examines whether HBM applies to all observations, we investigate the HBM model in detail with regard to three aspects. Specifically, we investigate how the sensitivity of the willingness to purchase the vaccine regarding the three explanatory variables changed during the three waves, and how the magnitude and sensitivity of the variables contributed to an increase in willingness. In addition, we examine whether the model applies to sub-samples classified by age, sex, income, or education. Furthermore, we examine whether the baseline model applies to the rule between individuals, although the baseline model focuses on the rule regarding within-individual changes. In Section 3, we explain how the perception of COVID-19, the willingness to pay for the vaccine, and risk attitude changed as the COVID-19 situation became more severe during the observation period. In Section 4, we present our empirical results. In Section 4.1, we show the results of the baseline estimation on how the willingness to purchase the vaccine depends on the perception of COVID-19 (the HBM) and risk attitude. In Section 4.2, Section 4.3 and Section 4.4, we present the results of the investigation of the extended models. Finally, Section 5 concludes the paper.

## 2. Data and Method

### 2.1. Survey

We collected panel data on how Japanese people perceive COVID-19 and determined their prevention behavior. We utilized Intage Inc. (Tokyo, Japan), a large internet survey company experienced in facilitating academic surveys. Before the survey, Intage Inc. asked 20,000 members from their respondent pool (screen survey) whether they will participate in the consecutive surveys for several months. The first wave was conducted from 13 March to 16 March. We aimed to collect 4000 responses and distributed the questionnaire to 7965 individuals who agreed in the screen survey, and ultimately received 4359 responses (a response rate of 54.3%). In the first wave we collected the respondents’ fixed attributes. The second wave was completed from 27 March and 30 March, and the questionnaire was distributed to all respondents from the first wave. We received 3495 responses as a result (a response rate of 80.2%). The third wave was conducted from 10 April and 13 April, in which the questionnaire was distributed to all respondents from the first wave, and we received 4013 responses (a response rate of 92.2%). The data are a representative sample of the residents in Japan with respect to sex, age (between 16–79 years), and region.

### 2.2. The COVID-19 Situation during the Three Waves of the Survey

#### 2.2.1. Situation during Wave 1

On 11 March, the World Health Organization declared COVID-19 a global pandemic. In Japan on 13 March when the first wave of our survey started, the Act on Special Measures against Pandemic Influenza was passed, which enabled the government to declare a state of emergency if and when required. On the same day, the USA declared a national emergency and planned to access up to $50 billion in federal funds. The number of positive cases in Japan increased to 675, more than seven times that on 20 February, while the number of deceased increased to 19. Though the situation in Japan worsened during these three weeks, the condition in the USA and many European countries became more serious: the numbers of positive cases and deceased were 17,660 and 1268 in Italy and 1264 and 36 in the USA, respectively (see Table 1 and Figure 1).

#### 2.2.2. Situation during Wave 2

The second wave of the survey started on 27 March, when the number of positive cases increased to 1387 (doubling over the course of 2 weeks), and the number of deaths increased to 46. The situation in European countries and the USA deteriorated rapidly, with the number of positive cases and deaths rising to 80,539 and 8165 in Italy and to 68,334 and 991 in the USA, respectively.

#### 2.2.3. Situation during Wave 3 and Thereafter

On 7 April, the Japanese Prime Minister declared a state of emergency in seven prefectures, including Tokyo and Osaka, and instructed citizens to avoid leaving their homes. The governors of these prefectures announced the closure of specific sectors, such as schools, museums, theaters, live performances, and hotels. This emergency was predicted to continue for a month. The third wave of our survey was conducted from 10 April to 13 April. On 10 April, the numbers of positive cases and deceased individuals were 5347 and 88 in Japan, 143,626 and 18,281 in Italy, and 425,889 and 14,665 in the USA, respectively. On 16 April, the regions in which emergency legislation applied, expanded to all prefectures. On 20 April, the number of positive cases and deaths were 10,751 and 171 in Japan and 723,605 and 34,203 in the USA, respectively. On 4 May, the emergency was extended until the end of May. However, it was lifted on 14 May, except for 8 out of 47 prefectures, and on 25 May it was completely withdrawn from Japan.

### 2.3. Questions and Variables

In this study, we use the answers to four questions asked throughout all three waves. The first question determines respondents’ willingness to purchase a hypothetical vaccine, which is assumed to be effective against COVID-19 and sold at 100,000 JPY (approximately US $1000). VACCINE is defined by the answer 1 (=Definitely not willing to purchase) to 5 (=Definitely willing to purchase). The second question determines respondents’ subjective probability of becoming infected with COVID-19 within one month. PROB is defined by the answer in terms of percentage (%). The third question enquires about possible symptoms if infected, in which the respondents are requested to select one from six scenarios: 1 (=minor influence) to 6 (=extremely serious symptoms). SEVERITY is defined according to the selected number. The fourth question requests the highest premium of insurance cover for the loss of 100,000 JPY (US $1000) with a 50% chance. Specifically, the respondents were questioned about the maximum amount they are willing to pay for the insurance and requested to select one option from the 11 cost options (i.e., insurance premium) displayed in ascending order. RA was defined as the selected numbers (between 1–11).

### 2.4. Model

#### 2.4.1. The Baseline Model

We aim to examine whether Japanese citizens rationally plan to prevent infection from COVID-19. Specifically, we investigate whether willingness to purchase a hypothetical vaccine depends on individuals’ perception of the seriousness level of COVID-19 and their risk attitude. Although this problem is scrutinized from a number of approaches in this study, the principal approach investigates whether a Japanese individual modified their willingness during the observed period following the rational model. Therefore, the problem is formalized as:

**Hypothesis** **1.***People**are more willing**to purchase the vaccine when they perceive COVID-19 as a more severe disease, when they consider a higher probability of infection from COVID-19, and when they become more risk averse*.

To test Hypothesis 1, we estimate the following equation with the fixed effect model, which analyzes within-individual variations:*VACCINE _i,t_* = *b*_0_ + *b*_1_*PROB*
*_i,t_* + *b*_2_*SEVERITY _i,t_* + *b*_3_*RA* + *b*_4_*WAVE2_t_* + *b*_5_*WAVE3_t_* + *e_i_* + *u_i,t_*(1)
where WAVE2 (WAVE3) is a dummy variable that takes 1 at the second (third) wave, and 0 otherwise. *e_i_* represents the fixed effect and *u_i,t_* is the random disturbance term. Hypothesis 1 is expressed as *b*_1_ > 0, *b*_2_ > 0, *b*_3_ > 0.

#### 2.4.2. Estimation of the Contribution of Each Variable to Willingness to Purchase a Vaccination

If Hypothesis 1 is confirmed, then the next inquiry is how the sensitivity of VACCINE on SEVERITY, PROB, and RA changed during the period. To identify this, we extend Equation (1) by incorporating the cross variables of the wave dummies with SEVERITY, PROB, and RA. For example, CRS2_SEVERITY is defined as the product of SEVERITY and WAVE2, and CRS3_PROB is defined as SEVERITY*WAVE3, and so forth. The coefficients on the cross terms represent the change in the sensitivity of the variables between waves 1 and 2, and 1 and 3, respectively. Furthermore, we estimate the contribution of each variable to the willingness to buy vaccines using the Blinder-Oaxaca decomposition, which is a standard method for estimating the contribution of each explanatory variable’s magnitude and sensitivity to the difference between the dependent variables [37,38,39,40]. In this study, we estimate the contribution of the change in magnitude and sensitivity of PROB, SENSITIVITY, and RA to the change in VACCINE between waves.

#### 2.4.3. Estimation of the Contribution of Each Variable to Willingness to Vaccination

Is the rational preventive behavior which is formalized as Hypothesis 1 supported by the data of sub-groups classified with some attributes? To examine this, we divide samples into male and female; high- and low-education groups; high- and low-income groups; and different age groups, and estimate Equation (1).

#### 2.4.4. Comparison between Individuals

In the baseline model and its extensions, we focus on how the willingness of individuals to purchase the vaccine changed during this period. This research is important because, using the knowledge acquired, public authorities can intervene and influence individuals to change their behavior. Another reason for its importance is that the fixed-effects estimation seems to control hidden confounders better than random-effects estimation and between-effects estimation. However, it is also interesting and important to explore how the willingness to pay for the vaccine differs among people because individuals may not necessarily follow the same rules, and knowledge about these differences is required to achieve agreement between individuals. Furthermore, it is interesting to investigate whether rules derived from observations of within- or between-individual variations are similar. Therefore, we make the following hypothesis:

**Hypothesis** **2.***An individual who perceives COVID-19 as a more severe disease, predicts a higher probability of infection, or is more risk-averse, is more willing to purchase the vaccine*.

To test Hypothesis 2, with each estimation method (BE, RE, and OLS), we regress VACCINE on PROB, SEVERITY, RA, and various control variables, which include sex, age, age-squared, educational level of graduation of respondents and of their parents, income and income-squared, religion, occupation, region of residence, residential city size, and individual characteristics such as altruism, trust, and optimism.

## 3. Descriptive Statistics of the Variables

In this subsection, we briefly explain the mean and standard deviation of the variables used in this study and how they changed over time. Mean and standard deviation of the variables of all waves and each wave are indicated in Table 2 and Table 3, respectively.

### 3.1. VACCINE

The mean of VACCINE significantly increased throughout the three waves. Nevertheless, the mean is smaller than the middle option even at wave 3, suggesting that most people are not willing to buy the vaccine. Means of willingness to buy vaccine for each wave indicate that about 30% answered 1 (=“definitely do not buy the vaccine”), while those who chose 4 or 5 (=”definitely buy”) was less than 10% throughout the waves (see Appendix A). On average, people do not want to pay 100,000 yen to obtain safety. The reason of this does not seem to be the high price of vaccine, since those with high income show a similar distribution. However, it also reveals that the willingness becomes larger during four weeks.

### 3.2. SEVERITY

The mean value of SEVERITY became larger over time. Those who considered “it takes from a week (=3)” or “one month (=4)” to recovery from the infection shared major portion, while those who chose 1 = “little influence” and 2 = “it takes a few days for recovery from infection” decreased from 25% at the 1st wave to 15% at the 3rd wave (Appendix A). However, those who considered COVID-19 a serious disease, which causes death or severe consequences, totaled only 15% even at the 3rd wave.

### 3.3. PROB

The mean of PROB increased from 21% at wave 1 to 30% at wave 3. The distribution of the answers aggregated by 10% each for each wave, reveals that a prominent change was the decline in the numbers who answered percentages less than 10%; it was 43% in the first wave and 20% in the third wave. Respondents answering less than 1% decreased from 25% to 6%. The increase in those who answered 50% increased from 20% to 29%. Nevertheless, even in wave 3, nearly half of the respondents answered less than equal to 30% (Appendix A).

### 3.4. Risk Attitude (RA)

Table 1 reveals that the mean of RA declined from 3.09 to 2.89 and then to 2.55 during the 4 weeks (Appendix A). These changes in means are significant at the 1% level. These results imply that people became more risk tolerant from wave 1 to wave 3.

### 3.5. Summary of the Four Variables

The mean of the magnitude of each variable for each wave i reveals that while the mean of VACCINE, SEVERITY, and PROB increased during the waves, RA decreased during the period (Appendix A). Coefficients of correlation between these variables are shown in Table 4. VACCINE is positively correlated with all the variables. The correlation with SEVERITY is the highest among them.

## 4. Results of the Estimation

### 4.1. The Results of the Baseline Model

Estimates of Equation (1) using the fixed effect model (FE) are presented in Table 5. In the first and second columns, the estimates of the specification considering only four main variables (KEY) are shown, while the third and fourth columns show the results of the full specification incorporating all the time-variant variables (FULL), which are three emotion variables, amount of information on COVID-19, achievement of four prevention measures, and six expectation variables for the spread of COVID-19 and income. In all columns, PROB, SEVERITY, and RA are positive and significant at the 1% level, supporting the HBM. Dummies representing wave 2 and wave 3 are significantly positive, while the latter is larger.

### 4.2. Which Variables Change Contributed to the Increase in the Willingness to Purchase the Vaccine

We present the estimation results of the extended model, which incorporates the cross-variables of the wave dummies with SEVERITY, PROB, and RA, respectively, into Equation (1). Table 6 reveals that while all the coefficients on the cross terms of PPROB and SENSITIVITY are positive, only those of SEVERITY with wave 2 are significant. Meanwhile, the sensitivity of RA increased between both waves. Coefficients on waves 2 and 3 become smaller compared to those in Table 5, indicating that a part of the increase in VACCINE is explained by the increase in the sensitivities of SEVERITY, PROB, and RA. The sensitivity of VACCINE on each variable at each wave, which is calculated from the estimates in Table 6 reveals that the sensitivity of SEVERITY increased from wave 1 to wave 2 and decreased from wave 2 to wave 3 (Appendix A). Sensitivity of PROB increased during all three waves, but only slightly. RA increased significantly throughout the three waves. These results suggest that while the contribution of the change in sensitivity of PROB and SEVERITY may be limited, RA may contribute throughout the change in sensitivity.

In Table 7, we present the results of Blinder-Oaxaca decomposition. In the left columns, the results of the change between waves 1 and 2 are shown. The first column indicates that the prediction of the magnitude of demeaned VACCINE based on our model is −0.217 at wave 1 and 0.0243 at wave 2. The increase in demeaned VACCINE between waves 1 and 2, namely 0.241, is to be decomposed to “Magnitude,” “Sensitivity,” and “Interaction” with the threefold decomposition. Here, Magnitude represents the part corresponding to the increase in the magnitude of the variables, namely (demeaned) SEVERITY, PROB, and RA given the magnitude of the sensitivity, namely the coefficients shown in Table 6. Sensitivity represents the part corresponding to the increase in sensitivity of the variables, that is, the coefficients on cross terms presented in Table 6, given the magnitude of these variables. Interaction represents the part corresponding to the simultaneous (i.e., the product of) increases in the magnitude and in the sensitivity.

“Total” in Table 7, which is the sum of the contributions of each variable, indicates that “Sensitivity” explains most of the contribution to the increase in VACCINE between waves 1 and 2. “Magnitude” explains only 10% of the increase and “Interaction” is minor. Comparison between the variables reveals that SEVERITY is more influential than PROB. RA affected through two opposite directions while “Magnitude” is negative because people became more risk tolerant, “Sensitivity” is positive, as shown in the positive coefficients of cross terms of risk aversion and wave dummies in Table 6. The aggregation of the three channels reveal that change in RA decreased VACCINE from wave 1 to wave 2. The right columns, which present the decomposition of the change from wave 2 to wave 3, indicate qualitatively similar results. The only difference is that the “Sensitivity” of RA became smaller. The change in RA, in total, decreased VACCINE from wave 2 to wave 3. Sensitivity of the constant term is large and significant in both changes, suggesting that the importance of “Sensitivity” comes from time-variant variables other than those included in the regression.

### 4.3. Difference in Prevention Behavior between Young, Middle, and Old Age Groups

In this subsection, we report the results for the different subgroups. We found that there were no systematic differences among the coefficients of SEVERITY, PROB, and RA between male and female, high- and low-education groups, and high- and low-income groups, except that the effect of RA is weaker for females than males and less significant for high- than low-education groups. Therefore, we report the results of the different age groups. We divided samples into those aged under 40 years, those aged between 40 to 59, and those aged over 60. The estimates with the FE are shown in Table 8. The coefficient on PROB is the largest for those aged over 60, for those aged between 40 to 59 it is only half, and for those aged under 40 it is one fourth and insignificant. Meanwhile, the coefficient on SEVERITY is significant for all age groups, and its magnitude is comparable between groups. RA is insignificant for those aged between 40 to 59. While SEVERITY is a strong predicter for VACCINE in any age group, PROB is not in those aged under 40. These results reveal that the sensitivity to the three key variables are different between age groups; Hypothesis 1 is supported only for those aged over 60.

### 4.4. Comparison between Individuals

In Table 9, we report the estimates of between-effects (BE), random-effects (RE), and ordinary least squares (OLS) to examine Hypothesis 2. Note that we show only the estimates of the key variables, even though all control variables are included in the estimations. The results of BE, RE, and OLS are quite similar and qualitatively similar to those of the FE shown in Table 5, suggesting that observations of within and between variations produce similar rules. In sum, by comparing individuals, those who perceive COVID-19 as a more severe disease, who consider a higher probability of infection, and who are more risk averse, are more willing to purchase the vaccine, supporting Hypothesis 2.

## 5. Conclusions

This study investigated how their willingness to purchase the (hypothetical) vaccine depends on their perceptions of COVID-19 and their risk attitudes. We conducted a large-scale (approximately 4000 respondents for each wave) panel survey three times between 13 March and 13 April and found the following:When respondents confronted the possibility of COVID-19 infection, their willingness to purchase a vaccine, subjective probability of becoming infected, and possible symptoms if infected increased over time, while they became more risk tolerant.By estimating the fixed effect model (FE), we found that a person is more willing to buy the vaccine when the person perceives COVID-19 as a more severe disease, when considering a higher probability of infection, and becomes more risk averse. The result provides evidence that people in Japan considered their preventive behavior rationally, as the HBM predicts.The HBM also applies to between people variation, that is, a person who perceives COVID-19 as a more severe disease, predicts a higher probability of infection, and is more risk averse are more willing to purchase the vaccine.We analyzed how the three factors contributed to the increase in willingness to purchase the vaccine. Blinder-Oaxaca decomposition reveals that the change in sensitivity contributed more than that of the magnitude of variables, the contribution of possible symptoms if infected is larger than subjective probability of becoming infected. The decrease in risk aversion (i.e., becoming more risk tolerant) during the 4-week period has two contradicting effects: while the decrease in Magnitude is negatively affected, the increase in sensitivity increased willingness to purchase vaccine. Since the effect of risk aversion is complex, an interpretation might be helpful. First, higher risk aversion promotes willingness to buy the vaccine shown in Table 5. On average, people became more risk tolerant over time (Table 1 and Appendix A), which decreased willingness to purchase vaccine (“Magnitude” in Table 7). Meanwhile, the sensitivity of willingness to purchase vaccine on risk aversion became larger (cross terms in Table 6), which promoted willingness to purchase vaccine (“Sensitivity” in Table 7).The HBM is supported for subgroups classified by sex, income, or education. However, it is not supported by young and middle-aged groups; the model applies only to the old-aged group who consider COVID-19 more seriously. The result is contrary to [28], which assessed the intention to be vaccinated for influenza and rubella. In the context of the COVID-19 pandemic, however, it is widely known that older people face more risks when infected, therefore, it is reasonable that they are more willing to take vaccines. For example, as of 6 January 2021, the cumulative number of deaths caused by COVID-19 among those aged over 60 was 95% of the total Japanese [41]. The Ministry of Labour, Health, and Welfare announced on 25 February 2021, “for the elderly and those with underlying illnesses, we will lead to earlier and more appropriate consultations, keeping in mind that they are likely to become severely ill” [42]. Therefore, the results obtained in this study are reasonable.Providing information about severe risks and susceptibility. However, the effect of the information was not necessarily clear-cut. In addition, they concluded that older individuals demonstrated vaccine hesitancy for both vaccinations, which seems opposite to the fact of the COVID-19 cases.

Furthermore, the importance of this study lies with the fact that it analyzes the data of individuals faced with the mega risk COVID-19. This enabled us to analyze the prevention behavior under the situation of a mega risk. Another accomplishment is the use of the panel data during the limited period of four weeks, which enabled us to assume that most attributes are fixed, while the situation concerning possible infections from COVID-19 changed radically. A difficulty of the HBM analyzed in this study is the endogeneity problem among key variables. First, willingness to purchase vaccine, subjective probability of becoming infected, possible symptoms if infected, and risk aversion may depend on hidden fixed confounders and time-variant factors during the observed period. The use of the FE and limited observation period minimized this problem. Another cause of the endogeneity is the reverse causality from prevention behavior to subjective probability of becoming infected and possible symptoms if infected. If a person takes the vaccine, subjective probability of becoming infected and possible symptoms if infected will be lowered. Questioning respondents about a hypothetical vaccine probably mitigates the reverse causality, if not perfects it. Given that previous literature on vaccines and HBM in the COVID-19 pandemic is scarce in Japan, we hope that the present study contributes to the accumulation of knowledge in the field.

Naturally, there are many limitations. Whether people truly make rational decisions and adopt rational behavior under the threat of COVID-19 and whether the prolonged request of self-control is effective, may be some of the most important problems to solve. This is more important in Japan, because the government is not allowed to restrict peoples’ and companies’ behavior, so that it only requests them to follow the guidelines, such as avoiding of going out or closure of operations. In this study, we found that citizens in Japan make rational decisions regarding willingness to purchase a hypothetical vaccine. However, the COVID-19 pandemic has been continuing for more than a year since the observation period. According to [32], which uses the data until 27 December 2020, the voluntary stay at home in response to the risk of infection and the government’s request to stay at home play an important role in the low proportion of infectious individuals and the large decrease in consumption in Japan, suggesting that Japanese people have been rational in their response to COVID-19. In addition, as of 19 November 2021, 75.8% of the total population completed the second dose [43], which is relatively high compared with other developed countries [44]. However, it is important to investigate in future work whether Japanese people have continued to be rational concerning vaccination.

Whether people rationally adopt true prevention behavior, such as avoiding going out and receiving government-approved vaccines also remains to be assessed in future research. Regarding the progress of vaccination in countries, easy and fast availability of the vaccine, low vaccination costs, having no serious side effects, having strong effectiveness, and availability of the vaccine at any facility are important factors. In addition, belief in religion also affects people’s willingness to be vaccinated [45]. Furthermore, there are three factors to consider when compelling people to adopt appropriate prevention behaviors. First, to fill the gap between plans (willingness) and behavior, people require a strong will to achieve the plan. Psychologists and behavioral economists argue that people have present bias and various misperceptions [46,47,48]. Therefore, if people fail to exercise preventive behavior such as “self-control regarding no outings,” we need to devise measures to promote appropriate behavior.

The second problem refers to the socially desirable level of preventive behavior that exceeds the desirable level of selfish individuals because of the externality of the infection. Discrepancies between the social optimum and the outcome of freedom may lead to limited freedom. Ref. [49] found that altruism plays an important role in promoting flu vaccines in Japan, suggesting that the externality problem might be mitigated if people are more considerate of others.

Third, although the free-rider problem will obstruct the internalization of the externality of the infection, Japanese individuals are characterized with more collectivistic rather than individualistic traits [50,51,52]. Economic experiments using public games found that Japanese people punish those who do not cooperate more than Americans [53,54]. These types of forces, such as spiteful behavior towards outliers, which was referred to as “Hachibu” in the past meaning “neglect from the community,” could discipline people to cooperate. Even though collectivism, companion consciousness within a small group, and spirit of community has been considered primitive in Japanese society, severe trials due to COVID-19 might stimulate reconsideration of communitarianism. If successful, request-based policies without enforcement by law could be effective in preventing the spread of the infection.

## Figures and Tables

**Figure 1 ijerph-18-12450-f001:**
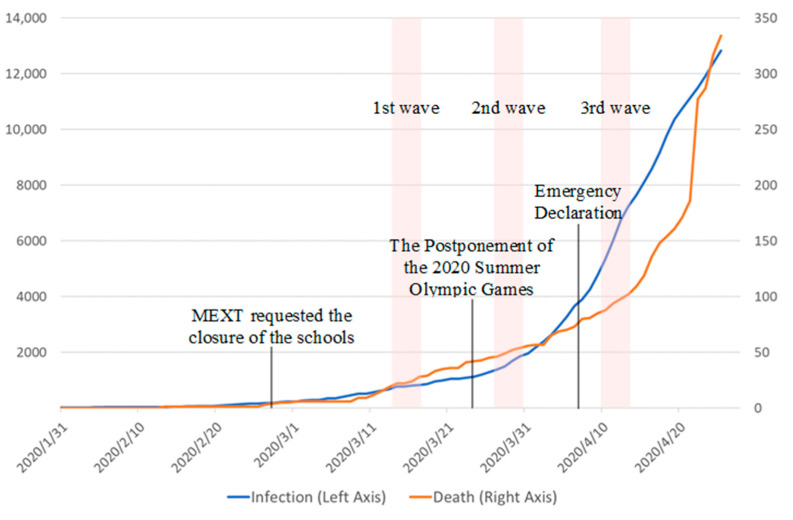
The number of the infected and the dead in Japan from 31 January to 26 April. Source: World Health Organization.

**Table 1 ijerph-18-12450-t001:** Descriptive statistics of the variables used in this paper: each wave.

	1st Wave: 13–16 March		2nd Wave: 27–30 March		3rd Wave: 10–13 April	
	# of Infected	# of Deaths	# of Infected	# of Deaths	# of Infected	# of Deaths
Japan	675	19	1387	46	5347	88
Italy	17,660	1268	80,539	8165	143,626	18,281
USA	1264	36	68,334	991	425,889	14,665
World	140,936	5362	512,701	23,447	1,524,151	92,940

**Table 2 ijerph-18-12450-t002:** Descriptive statistics of the variables used in this paper: All waves.

Variable	Observation	Mean	SD	Min	Max
VACCINE	11,867	2.295	1.121	1	5
SEVERITY	11,867	3.272	1.233	1	6
PROB	11,867	26.133	24.147	0	100
RA_MEGA	11,867	2.846	1.878	1	11

**Table 3 ijerph-18-12450-t003:** Descriptive statistics of the variables used in this paper: each wave.

	1st Wave: 13–16 March			2nd Wave: 27–30 March			3rd Wave: 10–13 April		
Variable	Observation	Mean	SD	Observation	Mean	SD	Observation	Mean	SD
VACCINE	4359	2.08	1.057	3495	2.312	1.114	4013	2.514	1.151
Increase %		NA			11.15			8.74	
SEVERITY	4359	3.044	1.202	3495	3.236	1.228	4013	3.55	1.215
Increase %		NA			6.31			9.70	
PROB	4359	21.7	24.983	3495	26.827	22.992	4013	30.34	23.381
increase %		NA			23.63			13.11	
RA	4359	3.09	2.021	3495	2.888	1.862	4013	2.546	1.68
increase %		NA			−6.54			−11.84	

Note: ‘increase %’ is the increase (%) of mean from the previous wave.

**Table 4 ijerph-18-12450-t004:** Correlation coefficients between variables.

	VACCINE	PROB	SEVERITY
PROB	0.0976	1	
*p*-value	0.0000		
SEVERITY	0.2251	0.1737	1
*p*-value	0.0000	0.0000	
RA	0.1637	0.0257	0.0365
*p*-value	0.0000	0.005	0.0001

Note: Correlation is calculated using all samples of three waves. The numbers in the lower rows are *p*-values of the significance of the correlation.

**Table 5 ijerph-18-12450-t005:** Estimates of Equation (1) with the fixed effect model (FE).

VARIABLES	KEY	FULL
PROB	0.00236 ***	0.00296 ***
	(0.000574)	(0.000766)
SEVERITY	0.0646 ***	0.0620 ***
	(0.0130)	(0.0171)
RA	0.0207 ***	0.0205 **
	(0.00674)	(0.00861)
WAVE2	0.228 ***	0.210 ***
	(0.0171)	(0.0238)
WAVE3	0.402 ***	0.353 ***
	(0.0184)	(0.0283)
Constant	1.760 ***	1.308 ***
	(0.0483)	(0.152)
OTHER VARIABLES	NO	YES
Observations	11,867	8098
R-squared	0.099	0.109
Number of individuals	4359	3615

Note: Dependent variable is VACCINE. Estimation method is FE. In columns heading with KEY, the estimates of Equation (1) is presented. In columns heading with FULL, the estimates of equation incorporating other time-variant variables, three emotion variables, amount of information on COVID-19, achievement of four prevention measures, and six expectation variables for spread of COVID-19 and income, are presented. Robust standard errors in parentheses; *** *p* < 0.01, ** *p* < 0.05.

**Table 6 ijerph-18-12450-t006:** Estimation results of Equation (1) incorporating cross terms.

VARIABLES	
PROB	0.00193 ***
	(0.000702)
CRS2_PROB	0.000531
	(0.000800)
CRS3_PROB	0.00121
	(0.000835)
SEVERITY	0.0503 ***
	(0.0160)
CRS2_SEVERITY	0.0348 **
	(0.0157)
CRS3_SEVERITY	0.00929
	(0.0162)
RA	−0.00578
	(0.00821)
CRS2_RA	0.0383 ***
	(0.00995)
CRS3_RA	0.0647 ***
	(0.0105)
WAVE2	−0.0102
	(0.0570)
WAVE3	0.164 ***
	(0.0596)
Constant	1.895 ***
	(0.0565)
Observations	11,867
R-squared	0.106
Number of individuals	4359

Note: Dependent variable is VACCINE. Estimation method is FE. Robust standard errors in parentheses; *** *p* < 0.01, ** *p* < 0.05.

**Table 7 ijerph-18-12450-t007:** Results of Oaxaca decomposition.

VARIABLES	Wave 1 vs. Wave 2	Magnitude	Sensitivity	Interaction		Wave 2 vs. Wave 3	Magnitude	Sensitivity	Interaction
PROB		0.0107 ***	−0.00117	0.00133			0.00874 ***	0.000327	0.00197
		(0.00315)	(0.00440)	(0.00501)			(0.00296)	(0.000666)	(0.00396)
SEVERITY		0.0154 ***	0.000853	−0.000740			0.0225 ***	0.000958	−0.00962
		(0.00301)	(0.00492)	(0.00426)			(0.00504)	(0.000741)	(0.00676)
RA		−0.00306 **	0.00212	−0.00172			−0.00851 ***	−0.000338	0.00256
		(0.00153)	(0.00275)	(0.00223)			(0.00303)	(0.000577)	(0.00424)
Constant			0.218 ***					0.172 ***	
			(0.0138)					(0.0144)	
Total		0.0230 ***	0.220 ***	−0.00113			0.0228 ***	0.173 ***	−0.00510
		(0.00453)	(0.0141)	(0.00684)			(0.00654)	(0.0146)	(0.00877)
VACCINE	0.0243 **				VACCINE	0.215 ***			
_wave2	(0.00953)				_wave3	(0.00930)			
VACCINE	−0.217 ***				VACCINE	0.0243 **			
_wave1	(0.00898)				_wave2	(0.00953)			
Increase in	0.241 ***				Increase in	0.190 ***			
VACCINE	(0.0131)				VACCINE	(0.0133)			
Observations	7854	7854	7854	7854		7508	7508	7508	7508

Note: Robust standard errors in parentheses; *** *p* < 0.01, ** *p* < 0.05.

**Table 8 ijerph-18-12450-t008:** Estimation results by age group.

VARIABLES	Under 40	Between 40 to 59	Over 60
PROB	0.000972	0.00190 **	0.00391 ***
	(0.00107)	(0.000958)	(0.000976)
SEVERITY	0.0743 ***	0.0595 ***	0.0529 **
	(0.0231)	(0.0220)	(0.0231)
RA	0.0349 ***	0.00670	0.0231 *
	(0.0122)	(0.0110)	(0.0120)
WAVE2	0.200 ***	0.210 ***	0.283 ***
	(0.0288)	(0.0293)	(0.0316)
WAVE3	0.391 ***	0.391 ***	0.432 ***
	(0.0315)	(0.0308)	(0.0341)
Constant	1.642 ***	1.730 ***	1.949 ***
	(0.0845)	(0.0780)	(0.0911)
Observations	3739	4220	3908
R-squared	0.093	0.096	0.111
Number of individuals	1397	1548	1428

Note: Dependent variable is VACCINE. Estimation method is FE. Robust standard errors in parentheses; *** *p* < 0.01, ** *p* < 0.05, * *p* < 0.1.

**Table 9 ijerph-18-12450-t009:** Estimation results of between-effects (BE), random-effects (RE), and ordinary least squares (OLS).

VARIABLES	BE	RE	OLS
PROB	0.00171 **	0.00230 ***	0.00202 ***
	(0.000700)	(0.000485)	(0.000463)
SEVERITY	0.124 ***	0.0963 ***	0.111 ***
	(0.0146)	(0.0105)	(0.00993)
RA	0.0886 ***	0.0501 ***	0.0766 ***
	(0.00872)	(0.00605)	(0.00609)
WAVE2	0.232 **	0.228 ***	0.224 ***
	(0.116)	(0.0183)	(0.0243)
WAVE3	0.261 **	0.392 ***	0.392 ***
	(0.118)	(0.0214)	(0.0268)
Constant	−0.0307	0.279	−0.00665
	(0.352)	(0.348)	(0.260)
Observations	10,558	10,558	10,558
R-squared	0.217		0.182
Number of individuals	4256	4256	

Note: BE is the between estimation, RE is the random effect model, and OLS is ordinary least squares. All the control variables, sex, age, age-squared, education of respondents and that of their parents, income and income-squared, religion, occupation, and region of residence, residence city size, and individual’s character like altruism, trust, and optimism, are included in the estimation, though they are not shown in this table. In parentheses robust standard errors are shown for RE and OLS, and ordinary standard errors for BE, respectively; *** *p* < 0.01, ** *p* < 0.05.

## Data Availability

The data presented in this study are available on request from the corresponding author.

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
