# Peer review of "The Willingness to Pay for a Hypothetical Vaccine for the Coronavirus Disease 2019 (COVID-19)"

_ijerph, 2021, doi:10.3390/ijerph182312450_

Round 1

Reviewer 1 Report

In this study, the authors have investigated the willingness to pay for a hypothetical COVID-19 vaccine, which helps analyzing how people make rational make decisions about prevention measures. The study seems to have been performed in an effective manner, with varied statistical methods explained in enough detail. The study also has a worthy motivation as it deals with a great challenge countries faced/have been facing regarding how to risk is perceived and avoided. The introduction provided a comprehensive literature review of the theme with studies from Chile, China, Ecuador and Malaysia. Authors also discussed contextual factors not related to the Health Belief Model, but still relevant for the COVID-19 discussion and perception of risk (unemployment, suicide, epidemiologic projections). They end the article listing important factors to be considered by decision and policy makers and health professionals when compelling people to adopt appropriate prevention behaviors, which is extremely useful beyond the COVID-19 context. The main criticism I have over this manuscript is regarding its structure, and there are improvements that can be done in this manuscript, mostly to refine the understanding of the study:

  • For Tables 1a and 1b, the explanation of the variables and how they were measures should be introduced before these tables are presented. The variables and what they represent is only explained later in Section 2.3.
  • For Table 1b, besides the absolute numbers, it could be useful to indicate how much the increases were in percentage.
  • Section 2 (Data and Methods) explains how the survey was conducted and what the variables represent. However, in Section 4, other analysis were conducted that were not previously explained in Section 2 (such as the analysis of subgroups in Section 4.3)
  • Section 4 (Results of the Estimation) is convoluted: methods, results, and implications of results are all presented together, making it harder to identify the main findings for each type of analysis.
  • Section 5 (Conclusion) to use the name of the variable (VACCINE, SEVERITY, and PROB) may not be the best choice to convey the findings of the study in an intelligible manner.

My suggestion is to restructure the methods/results/conclusion sections to make it clearer for the reader what to expect, what is being analyzed, and what are the implications of the findings of the statistical analysis.

The discussion/conclusion of findings is also missing on how those connect to the knowledge acquired in other studies on this field. For example, in line 79 authors discuss that a previous study found vaccine hesitancy among older individuals in Japan. In sections 4 and 5 authors discuss that the HBM model applies only to the old-aged groups who consider COVID-19 more seriously. What are the implications of this finding regarding the vaccine hesitancy?

Finally, as the survey was conducted in 2020, authors may consider if it is relevant to also briefly mention in the discussion how was been the vaccine uptake in Japan by now.

As a minor subject, I present indicated some specific suggestions below:

Line 57: Numerous studies have done about vaccination against COVID-19 and health belief model in the COVID-19 pandemic. (missing “been” between “have” and “done”)

Line 115: We conducted a panel survey from March 13 to April 13 (missing “2020” in the end)

Line 170: The number of the in fected and the dead in Japan (a space in the middle of “Infected”)

Line 233: pay100 thousand (no space between “pay” and “100”)

Line 234: The reason of this does not seem the high price of vaccine (missing “to be” between “seem” and “the high”

Table 2: Put the p-values in italic

Author Response

Reply to the reviewer 1’s comments

Thank you very much for your insightful and detailed comments. We have substantially revised the previous version of the manuscript following your specifications. We hope that the revised manuscript is satisfactory. In the text provided below, your comment appears first in blue, and our replies follow.

Comments and Suggestions for Authors:

 In this study, the authors have investigated the willingness to pay for a hypothetical COVID-19 vaccine, which helps analyzing how people make rational make decisions about prevention measures. The study seems to have been performed in an effective manner, with varied statistical methods explained in enough detail. The study also has a worthy motivation as it deals with a great challenge countries faced/have been facing regarding how to risk is perceived and avoided. The introduction provided a comprehensive literature review of the theme with studies from Chile, China, Ecuador and Malaysia. Authors also discussed contextual factors not related to the Health Belief Model, but still relevant for the COVID-19 discussion and perception of risk (unemployment, suicide, epidemiologic projections). They end the article listing important factors to be considered by decision and policy makers and health professionals when compelling people to adopt appropriate prevention behaviors, which is extremely useful beyond the COVID-19 context. The main criticism I have over this manuscript is regarding its structure, and there are improvements that can be done in this manuscript, mostly to refine the understanding of the study:

Our reply 1: Thank you for your evaluation. We have extensively revised the structure of our manuscript following your specifications, as stated in reply 4.

For Tables 1a and 1b, the explanation of the variables and how they were measures should be introduced before these tables are presented. The variables and what they represent is only explained later in Section 2.3.

Our reply 2: Thank you for your careful review. To address this comment, we revised the previous version as follows. First, we noticed that the problem emerged mainly because Tables 1a and 1b in the previous version were inserted inappropriately. Only the upper part of Table 1b should be included in subsection 2-2, while the other parts detailing descriptive statistics should be included in Section 3. Therefore, we renamed the upper part of Table 1b as Table 1a, and the lower part of Table 1b as new Tables 1, 2a, and 2b, respectively, and placed them in appropriate locations.

For Table 1b, besides the absolute numbers, it could be useful to indicate how much the increases were in percentage.

Our reply 3: We added the increase/decrease of the mean values from the previous wave (%) to the new Table 2b.

Section 2 (Data and Methods) explains how the survey was conducted and what the variables represent. However, in Section 4, other analysis were conducted that were not previously explained in Section 2 (such as the analysis of subgroups in Section 4.3)

Section 4 (Results of the Estimation) is convoluted: methods, results, and implications of results are all presented together, making it harder to identify the main findings for each type of analysis.

Our reply 4: Thank you for your comment. We moved the paragraphs which explain the purposes and methods in subsections 4.2-4.4, to the new subsections 2.4.2-2.4.4.

Section 5 (Conclusion) to use the name of the variable (VACCINE, SEVERITY, and PROB) may not be the best choice to convey the findings of the study in an intelligible manner.

Our reply 5: Following your comment, we replaced VACCINE, SEVERITY, and PROB with willingness to purchase the vaccine, subjective probability of becoming infected, and possible symptoms if infected, respectively, in the Conclusion section. We hope that our meaning is conveyed intelligibly.

My suggestion is to restructure the methods/results/conclusion sections to make it clearer for the reader what to expect, what is being analyzed, and what are the implications of the findings of the statistical analysis.

Our reply 6: Please refer to reply 4 We hope that revising the manuscript has improved readability.

The discussion/conclusion of findings is also missing on how those connect to the knowledge acquired in other studies on this field. For example, in line 79 authors discuss that a previous study found vaccine hesitancy among older individuals in Japan. In sections 4 and 5 authors discuss that the HBM model applies only to the old-aged groups who consider COVID-19 more seriously. What are the implications of this finding regarding the vaccine hesitancy?

Our reply 7: Thank you for your insightful comments. We have substantially revised the Conclusion section. Regarding vaccine hesitancy among older individuals, we added the following: 

The result is contrary to [28], which assessed the intention to be vaccinated for influenza and rubella. In the context of the COVID-19 pandemic, however, it is widely known that older people face more risks when infected, therefore, it is reasonable that they are more willing to take vaccines. For example, as of January 6, 2021, the cumulative number of deaths caused by COVID-19 among those aged over 60 was 95% of the total Japanese [41]. The Ministry of Labour, Health, and Welfare announced on February 25, 2021, “for the elderly and those with underlying illnesses, we will lead to earlier and more appropriate consultations, keeping in mind that they are likely to become severely ill” [42]. Therefore, the results obtained in this study are reasonable.

In addition, we also added the following paragraph in the conclusion section:

Whether people rationally adopt true prevention behavior, such as avoiding going out and receiving government-approved vaccines also remains to be assessed in future research. Regarding the progress of vaccination in countries, easy and fast availability of the vaccine, low vaccination costs, having no serious side effects, having strong effectiveness, and availability of the vaccine at any facility are important factors. In addition, belief in religion also affects people’s willingness to get vaccinated [45].

Finally, as the survey was conducted in 2020, authors may consider if it is relevant to also briefly mention in the discussion how was been the vaccine uptake in Japan by now.

Our reply 7: Thank you for your insightful comment. We added the following paragraph in the conclusion section:

However, COVID-19 has been ongoing for more than a year after the observation period. According to [32], which uses the data until December 27, 2020, the voluntary stay at home in response to the risk of infection and the government’s request to stay at home play an important role in the low proportion of infectious individuals and the large decrease in consumption in Japan, suggesting that Japanese people have been rational in their response to COVID-19. In addition, as of November 19, 2021, 75.8% of the total population completed the second dose [43], which is relatively high compared with other developed countries [44]. However, it is important to investigate in future work whether Japanese people have continued to be rational concerning vaccination.

As a minor subject, I present indicated some specific suggestions below:

Line 57: Numerous studies have done about vaccination against COVID-19 and health belief model in the COVID-19 pandemic. (missing “been” between “have” and “done”)

Line 115: We conducted a panel survey from March 13 to April 13 (missing “2020” in the end)

Line 170: The number of the in fected and the dead in Japan (a space in the middle of “Infected”)

Line 233: pay100 thousand (no space between “pay” and “100”)

Line 234: The reason of this does not seem the high price of vaccine (missing “to be” between “seem” and “the high”

Table 2: Put the p-values in italic

Our reply 8: Thank you very much for your careful review. We have corrected all the errors that you have pointed out.

Reviewer 2 Report

The authors of the work presented ,, The willingness to pay for a hypothetical vaccine for the COVID-19. The study was well designed, and the authors are aware of the study's numerous limitations. I would suggest adding information on willingness to vaccinate in other countries and highlighting the practical aspects of the work. 

Author Response

Reply to the reviewer 2’s Comments and Suggestions

Reviewer 2’s Comments and Suggestions:

The authors of the work presented ,, The willingness to pay for a hypothetical vaccine for the COVID-19. The study was well designed, and the authors are aware of the study's numerous limitations. I would suggest adding information on willingness to vaccinate in other countries and highlighting the practical aspects of the work.

Our reply:

Thank you very much for your insightful comment. To address your suggestion, we added the following paragraphs in the conclusion section. We hope that the revision is satisfactory to you.

However, COVID-19 has been ongoing for more than a year after the observation period. According to [32], which uses the data until December 27, 2020, the voluntary stay at home in response to the risk of infection and the government’s request to stay at home play an important role in the low proportion of infectious individuals and the large decrease in consumption in Japan, suggesting that Japanese people have been rational in their response to COVID-19. In addition, as of November 19, 2021, 75.8% of the total population completed the second dose [43], which is relatively high compared with other developed countries [44]. However, it is important to investigate in future work whether Japanese people have continued to be rational concerning vaccination.

Whether people rationally adopt true prevention behavior, such as avoiding going out and receiving government-approved vaccines also remains to be assessed in future research. Regarding the progress of vaccination in countries, easy and fast availability of the vaccine, low vaccination costs, having no serious side effects, having strong effectiveness, and availability of the vaccine at any facility are important factors. In addition, belief in religion also affects people’s willingness to get vaccinated [45].
